# Utilization of biochar derived from industrial hemp stalks with various cooling methods for asphalt binder modification

İbrahim Aslan[1]*, Funda Tasdemir[2], Yuksel Tasdemir[3]

1 Department of Material and Energy, Hemp Research Institute, Yozgat Bozok University, Yozgat, Türkiye,
2 Department of Mathematics, Faculty of Arts and Science, Yozgat Bozok University, Yozgat, Türkiye,
3 Department of Civil Engineering, Faculty of Engineering and Architecture, Yozgat Bozok University, Yozgat, Türkiye

* ibrahim.aslan@bozok.edu.tr

## Abstract

The study aimed to identify the optimum pyrolysis temperature and cooling conditions for producing biochar from industrial hemp stalks to enhance asphalt binder performance, while also assessing the potential of industrial hemp stalks as a viable alternative in the pavement sector by evaluating the temperature sensitivity and the high and low temperature performance of biochar-modified binders. The asphalt binders were subjected to conventional and performance tests to assess their properties. Penetration and softening point values of neat and biochar modified binders were determined and the penetration index of binders was calculated. The temperature susceptibility, permanent deformation, aging and fatigue performances of neat and biochar modified asphalt binders were determined by the dynamic shear rheometer test. Bending beam rheometer equipment was used to determine the low temperature performance of asphalt binders. The experimental results showed that biochar obtained from industrial hemp stalk improved aging resistance, reduced the temperature susceptibility, increased high-temperature performance grade, and so developed permanent deformation resistance of the binder. The optimal production conditions for biochar, utilized in the modification of asphalt binders to achieve better rutting resistance, have been identified as a combination of a pyrolysis temperature of 300°C and rapid cooling. The low-temperature performance grade of biochar-modified asphalt binders decreased by one grade, while the high-temperature performance grade increased by at least one grade. The addition of biochar was shown to have a negative impact on the binder's low-temperature performance. Based on these findings, it was concluded that the biochar additive, derived from industrial hemp stalk, is particularly well-suited for regions characterized by hot climates.

**Data availability statement:** All relevant data are within the paper and its Supporting Information files.

**Funding:** This work was supported by Yozgat Bozok University with project code FKA-2022-1005 and the Scientific and Technological Research Council of Türkiye (TUBITAK) with project code 122M886. The funders had no role in study design, data collection and analysis, decision to publish, or preparation of the manuscript.

**Competing interests:** The authors have declared that no competing interests exist.

## 1. Introduction

Rapid population growth and advances in automotive technology lead to increase road traffic loads. These developments cause flexible road pavements to deteriorate in a short time and shorten their service life. There are main three types of asphalt pavement deterioration: fatigue cracking, thermal cracking and rutting. These deterioration cause to accidents causing fatalities, injuries, and property damage and have a considerable impact on the nation's economy. In order to prevent deterioration, it is of great importance to improve the properties of asphalt binders used in road construction and to increase the performance of roads by developing innovative additives. Various additives are currently employed to enhance binder properties and improve pavement performance. While studies have investigated the use of Styrene Butadiene Styrene (SBS), Styrene Butadiene Rubber (SBR), Ethylene Glycidyl Acrylate (EGA) terpolymer, waste plastics, waste tire rubber, fibers and waste fibers and warm mix asphalt additives for bitumen modification, there is also a growing interest in examining the effects of biomass based pyrolytic liquid products and biochar on binders [1,2].

Biomass as a renewable energy source refers to all animals, plants, and microorganisms that can absorb carbon dioxide for photosynthesis to produce various organic substances [3–5]. Biomass can be converted into solid, liquid and gas forms by chemical conversion techniques and used as an energy source or additive material. Pyrolysis, the most commonly used and extensively studied process for biochar production, involves the thermal decomposition of biomass at elevated temperatures in an oxygen-free environment [6–8]. During pyrolysis, the biomass undergoes thermal decomposition, releasing volatile compounds and leaving behind a solid residue known as biochar. Biochar can be produced from a wide range of biomass, including agricultural waste, forestry waste, and municipal and industrial organic waste. The yield and properties of the biochar depend on various factors such as the type of biomass, the heating rate, pyrolysis temperature, and the residence time [9]. There are three different pyrolysis processes: slow, fast and flash pyrolysis. Slow pyrolysis yields more biochar products at low temperature and low heating rate, while fast pyrolysis yields higher bio-oil product at high temperature such and high heating rate. In flash pyrolysis, the heating rate is higher than in fast pyrolysis and the processing time is quite short [10]. In slow pyrolysis, characterized by low reaction temperatures and heating rates, the biochar yield is maximized, resulting in the production of high-quality biochar [11,12]. Most of the previous laboratory scale research on slow pyrolysis investigated the effect of type of biomass, pyrolysis temperature, heating rate, and residence time, as the main operating variables that affected the biochar yield and properties [8,13–19]. Biochar derived from different biomass sources has been increasingly utilized in recent years. The primary applications of biochars currently lie within the agricultural sector [20–24]. In the construction industry, biochars have predominantly been investigated as a partial substitute for cement in mortar or concrete [25–30]. Biochars are also used in the modification of asphalt binders and below is a summary of studies on the use of biochar in asphalt binders.

According to Zhou et al. [31,32], while the characteristics of biochar can vary based on the pyrolysis conditions and the biomass source, the biochar presented in the research demonstrated to be an efficient modifier to improve the rheological properties of the asphalt binder and its performance against major pavement distresses.

Kumar et al. [33], demonstrated that both tire pyrolytic char and waste plastic pyrolytic char enhanced the high-temperature performance and fatigue cracking resistance of asphalt binders. Notably, the tire pyrolytic char-modified binder exhibited superior aging resistance compared to unmodified binders.

The results of the study by Dong et al. [34], showed that biochar tends to reduce low temperature performance while improving the aging resistance of asphalt binder. In a study using biochar obtained from various biomasses, Celoglu et al. [35], stated that biochar increased the stiffness and performance grade of the asphalt binder at high temperatures. Kumar et al. [36], conducted a study on asphalt binders using biochar obtained from the seed husk of the Mesua Ferrea plant and concluded that biochar increases the viscosity and permanent deformation resistance of the asphalt binder while decreasing the aging susceptibility. In a study conducted by Walters et al. [37], biochar obtained from pig manure as bio-mass was used as a asphalt binder additive and it was reported that biochar improved the aging resistance of the asphalt binder. Gan and Zhang [38], confirmed that the addition of crop straw biochar significantly enhanced the high temperature performance of asphalt, while it negatively impacted its low temperature performance. According to previous studies, it is recommended to use biochar particle size below 0.075 mm. In addition, the optimal level for biochar content ranging from 2% to 20% in studies is below 10% [8].

Factors affecting the yield and properties of biochar have also been taken into account in previous studies on biochar-modified asphalt binders and their mixtures [8]. No detailed study has been conducted on the cooling of biochar obtained after pyrolysis. In some studies, biochar was extracted from the steel reactor only after it had cooled to room temperature following the pyrolysis process, a procedure called as slow cooling, whereas other studies did not specify the cooling method. In this study, after pyrolysis, the hot biochar was immediately removed from the steel reactor, rapidly cooled by immersing it in water at 10°C, and then dried. This process was termed rapid cooling. In the study, biochar obtained by both methods was used in the experiments. The aim of the study was to determine the optimum pyrolysis temperature and cooling conditions for producing biochar from industrial hemp stalks to enhance asphalt binder performance. Additionally, the study evaluated the potential of industrial hemp stalks as a viable alternative in the pavement sector by analyzing the temperature sensitivity and high- and low-temperature performance of the binder with biochar incorporation. Although there are studies on biochar production from industrial hemp stalk for various purposes [39–42], as far as the authors are aware, no detailed research has been conducted on the performance properties of the asphalt binder incorporating industrial hemp stalk biochar.

## 2. Material

### 2.1. Neat asphalt binder

Neat asphalt binder with B50/70 penetration grade used in this study was sourced from TUPRAS Kırıkkale petroleum refinery. The properties of neat asphalt binder are given in Table 1.

### 2.2. Industrial hemp

Industrial hemp is an annual herbaceous plant of great value as a versatile industrial crop, with its fibers, roots, seeds, stems, leaves, and flowers being utilized across numerous sectors. The amounts of chemical constituents of industrial hemp depend on the cultivar, geographical origin, and time of harvest [43,44].

In the study, the industrial hemp plant grown and harvested by Yozgat Bozok University, Hemp Research Institute was used as a biomass source. Industrial hemp stalks with fibers removed were prepared for the pyrolysis process by grinding to a size smaller than 0.25 mm. Industrial hemp parts of 10–30 mm in size were also used for elemental analysis.

**Table 1. Properties of neat asphalt binder.**

| Properties | Standard | B 50/70 |
|---|---|---|
| Penetration (dmm) | EN 1426 | 51.2 |
| Softening Point (°C) | EN 1427 | 48 |
| Penetration Index (PI) | | −1.66 |
| Specific Gravity | EN 15326 | 1.03 |
| Viscosity (Pa.s, 135 °C) | EN 13302 | 0.4375 |
| Viscosity (Pa.s, 165 °C) | EN 13302 | 0.1 |

## 2.3. Biochar

The physical, chemical, and mechanical properties of biochar obtained by thermochemical degradation of biomass vary depending on the properties and production conditions of biomass [45,46]. In this study, the pyrolysis of industrial hemp stalks was conducted at three temperatures (300°C, 450°C, and 600°C) under two cooling methods (slow and rapid cooling) to identify the optimal production temperature and cooling conditions. To maximize biochar yield, the slow pyrolysis process was employed, with temperatures chosen within the typical range of 300–700°C for this method [6–8,47]. The pyrolysis system employed for biochar production illustrated in Fig 1. Allowing the biochar to cool to room temperature within the steel reactor after the pyrolysis process is referred to as slow cooling, while rapidly cooling the biochar by immersing it in cold water immediately after its removal from the steel reactor is referred to as rapid cooling. The processes of the rapid cooling and slow cooling are schematically shown in Fig 2. After completing all the processes, the biochar was sieved through a 75 μm sieve to be utilized in the modification of the asphalt binder. Throughout the article, biochar obtained by slow and rapid cooling processes are referred to as S and R, respectively. The industrial hemp products used in the study are shown in Fig 3.

## 2.4. Preparation of modified asphalt binders

Based on previous studies reporting that biochar content in asphalt binder ranged from 2% to 20% by weight, biochar concentrations of 0, 5, 10 and 15% were used in this study [8]. The procedure for the modification was carried out with a high-shear mixer at a rotational speed of 1000 rpm for a period of thirty minutes at a temperature of 150 °C. Asphalt binders were named according to biochar production parameters and biochar contents. For example, the abbreviation '300R5' represents a biochar-modified asphalt binder produced at 300°C with rapid cooling, incorporating 5% biochar.

## 3. Experimental procedure

### 3.1. Elemental and scanning electron microscope analysis of biochar

Elemental analysis was conducted at Inönü University Scientific and Technological Research Centre to determine the carbon (C), hydrogen (H), and nitrogen (N) ratios in raw industrial hemp stalks and the biochars produced through pyrolysis.

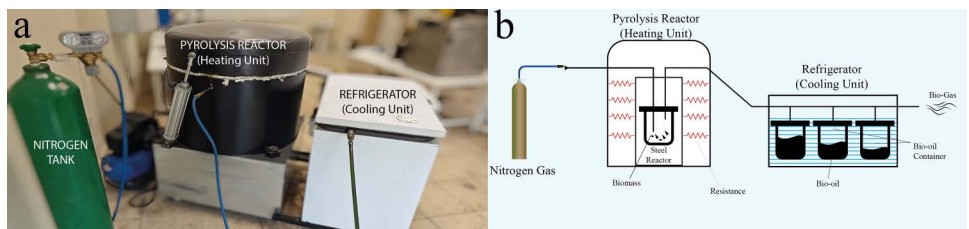

**Fig 1. (a) Pyrolysis system and (b) schematic illustration.**

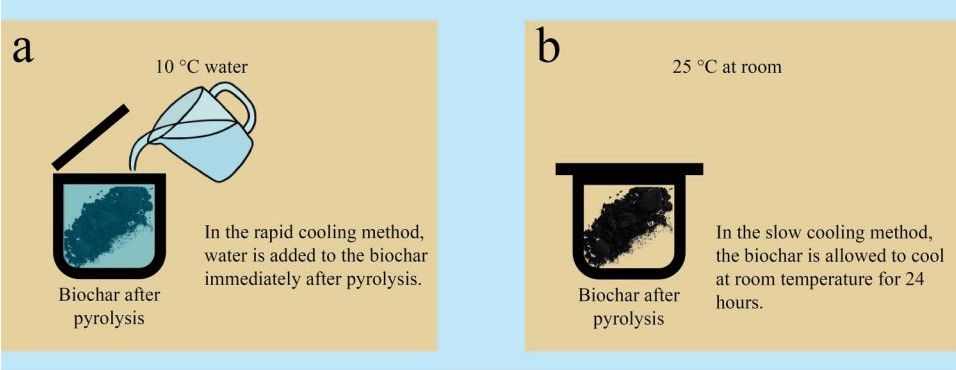

**Fig 2. (a) Rapid cooling process and (b) slow cooling process.**

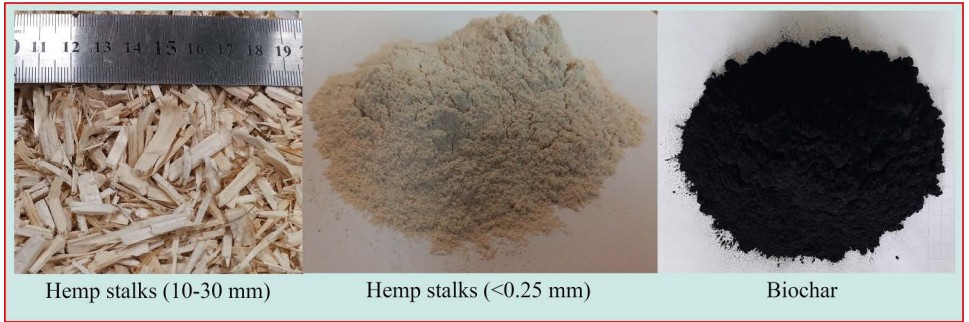

Hemp stalks (10-30 mm)   Hemp stalks (<0.25 mm)   Biochar

**Fig 3. Industrial hemp products.**

The surface morphologies of biochars obtained by rapid and slow cooling methods were examined using scanning electron microscopy (SEM). SEM analyses were carried out at the Science and Technology Application and Research Centre of Yozgat Bozok University.

### 3.2. Conventional asphalt binder tests

The conventional tests for asphalt binders, such as the softening point (EN 1427) and penetration at 25 °C (EN 1426), were performed to evaluate the material's properties. The penetration index (PI) values of the asphalt binders were calculated using their penetration and softening point measurements. This value is inversely proportional to the temperature susceptibility of asphalt binders and provides information about their temperature susceptibility. As asphalt binder's PI values decrease, their temperature susceptibility increases. The Penetration Index (PI) was calculated using the formula developed by Pfeiffer and Van Doormal [48].

$$PI = \frac{1952 - 500 x log(Pen) - 20 x SP}{50 x log(Pen) - SP - 120}$$

(1)

where; Pen is the penetration at 25 °C and SP is the softening point value.

## 3.3. Rotational viscometer test

The rotational viscometer (RV) test is performed to determine the high-temperature viscosity characteristics of asphalt binders. The viscosity values of asphalt binders at high temperatures are measured to determine whether the asphalt binders are sufficiently fluid for mixing. In the test, viscosity values are obtained by the resistance to rotation of a shaft rotating at 20 rpm in the asphalt binder [49,50].

In this study, the viscosity values of each asphalt binder group were measured at 10°C intervals within the temperature range of 130°C to 170°C to examine the effect of biochar modification on the temperature-dependent viscosity of bitumen.

## 3.4. Aging process of asphalt binder

The aging of the asphalt binders was conducted using a combination of the Rolling Thin Film Oven Test (RTFOT, EN 12607-1) and a Pressure Aging Vessel (PAV, EN 14769). RTFOT represents the short-term aging of asphalt binders. In other words, the RTFOT represents the aging of asphalt binder due to temperature during the manufacturing and construction of asphalt mixtures. In the RTFOT, $35 \pm 0.5$ g of asphalt binder was placed in each of 8 specially designed bottles. The test was carried out at 163 °C for 85 minutes and the bottles were sprayed with compressed air at $4000 \pm 200$ ml/min during the experiment. The Pressure Aging Vessel (PAV) test is a common laboratory test used to simulate the long-term aging that occurs during the service life of an asphalt pavement. In the PAV test, the asphalt binder is first short term aged using the rolling thin film oven (RTFO) test. The sample is then placed in a PAV vessel and subjected to a temperature of 100 °C and a pressure of 2070 kPa for 20 hours.

## 3.5. DSR test

The DSR enables the measurement of the complex modulus (G*) and the phase angle ($\delta$) of asphalt binders at high temperatures. G*, an indicator of the resistance of asphalt binders to shear deformations in the viscoelastic region, and $\delta$ can be used to evaluate the rutting and fatigue behaviour of asphalt mixtures. The rutting resistance of asphalt binders was evaluated using the SUPERPAVE rutting parameter, G*/sin$\delta$. In the SUPERPAVE asphalt binder specification, the G*/sin$\delta$ value for unaged asphalt binders is limited to a minimum value of 1.0 kPa [51]. The value of G*.sin$\delta$ represents the stiffness at moderate temperatures and is an indicator of fatigue resistance. The maximum G*.sin$\delta$ value of PAV-aged asphalt binders is limited to 5000 kPa [52].

Complex modulus aging index (CAI), phase angle aging index (PAI) and rheological aging index (RAI) were determined using G* and $\delta$ of unaged and RTFOT aged asphalt binders. CAI, PAI and RAI were used to evaluate the aging susceptibility of the asphalt binders. The formulas used to calculate the aging indexes are presented in Table 2.

## 3.6. Multiple Stress Creep Recovery (MSCR) test

In order to ascertain the high temperature and rutting behaviour of asphalt binders, an MSCR test was also conducted at 64 °C in accordance with the AASHTO T350 standard [55]. Percentage recovery R, permanent creep compliance Jnr, and stress susceptibility $Jnr_{diff}$ values were determined as a result of the test. While the percentage recovery value gives information about the elastic behaviour of the material, the material with larger R-value shows more elastic behaviour [56]. The

**Table 2. Aging indexes [53,54].**

| Aging indexes | Calculation method |
| --- | --- |
| CAI | $CAI = G^*_{aged}/G^*_{unaged}$ |
| PAI | $PAI = \delta_{aged}/\delta_{unaged}$ |
| RAI | $RAI = (G^*/\sin\delta)_{aged}/(G^*/\sin\delta)_{unaged}$ |

$Jnr_{(3.2)}$ value mainly characterizes the rutting resistance and the asphalt binder with a low $J_{nr}$ value shows better behaviour against permanent deformations [57,58].

### 3.7. Bending Beam Rheometer (BBR) test

BBR test was performed according to the EN 14771 to determine the low-temperature stiffness of asphalt binders. Test specimens with dimensions of 6.25x12.7x125 mm were loaded with a load of 980±5 mN for 240 s. The creep stiffness (St) and creep rates (m-value) of asphalt binders are determined using the bending stress and strain at a loading time of 60 seconds. The creep stiffness shows the resistance of the asphalt binder against creep stresses and the creep rate shows the stiffness change in the asphalt binder during the loading period. These parameters represent the resistance of asphalt mixtures to low-temperature cracking.

## 4. Results and discussion

### 4.1. Elemental and SEM analysis results

Table 3 presents the elemental analysis results of biochars produced from industrial hemp stalks of varying sizes under different pyrolysis conditions. The elemental analysis results indicated that raw hemp stalk with a particle size smaller than 0.25 mm contained 42.16% carbon, 5.995% hydrogen, and 0.257% nitrogen, while raw hemp stalk with a particle size of 10–30 mm contained 45.63% carbon, 6.435% hydrogen, and 0.36% nitrogen. Our findings consistent with the data reported in the literature, emphasizing that the carbon content of raw hemp biomass ranges from 45% to 49%, hydrogen from 5.6% to 6.2%, and nitrogen from 0.3% to 1.3% [41,59,60]. Table 3 shows that the increase in the carbon amount of biochar obtained from the pyrolysis of small-sized biomass is high. The carbon content of hemp stalk biochar with a particle size smaller than 0.25 mm was found to range from 69.73% to 95.93%, while that of biochar with a particle size of 10–30 mm ranged from 65.5% to 83.85%, depending on the pyrolysis temperature and the cooling conditions. Changing the pyrolysis temperature and cooling conditions did not lead to a meaningful change in the observed increase in carbon. Hydrogen content exhibits an inverse relationship with temperature, decreasing as pyrolysis progresses. For small particles, hydrogen content drops from 3.884% at 300°C to 1.62% at 600°C under rapid cooling conditions. This reduction is attributed to the volatilization of hydrogen-containing compounds and the progressive formation of aromatic structures,

**Table 3. Elemental analysis results.**

| Biomass size, mm | Pyrolysis temperature, °C | Cooling condition | Content, % | | |
| --- | --- | --- | --- | --- | --- |
| | | | C | H | N |
| <0.25 | Raw hemp stalk | Raw hemp stalk | 42.16 | 5.995 | 0.257 |
| <0.25 | 300 | Rapid | 86.23 | 3.884 | 0.079 |
| <0.25 | 450 | Rapid | 95.93 | 2.885 | 0.081 |
| <0.25 | 600 | Rapid | 69.73 | 1.62 | 0.131 |
| <0.25 | 300 | Slow | 91.18 | 4.363 | 0.116 |
| <0.25 | 450 | Slow | 71.9 | 2.666 | – |
| <0.25 | 600 | Slow | 69.97 | 1.459 | – |
| 10-30 | Raw hemp stalk | Raw hemp stalk | 45.63 | 6.435 | 0.36 |
| 10-30 | 300 | Rapid | 69.64 | 3.841 | 0.267 |
| 10-30 | 450 | Rapid | 73.75 | 2.42 | 0.11 |
| 10-30 | 600 | Rapid | 77.72 | 1.652 | 0.251 |
| 10-30 | 300 | Slow | 83.85 | 3.576 | 0.397 |
| 10-30 | 450 | Slow | 65.5 | 2.308 | – |
| 10-30 | 600 | Slow | 80.57 | 1.188 | 0.094 |

which lead to lower H/C ratios and increased thermal stability of the biochar. Similarly, nitrogen content decreases with increasing temperature as nitrogenous compounds volatilize into gaseous emissions such as $NH_3$ and NOx.

The SEM images of biochar obtained from hemp stalk with a size of less than 0.25 mm at 300 °C by rapid and slow cooling conditions are presented in Fig 4.

Fig 4 illustrates that the biochars obtained using the rapid cooling method (b) exhibit reduced particle sizes in comparison to those obtained from the slow cooling method (a). This indicates that biochars produced by the rapid cooling method may possess a high specific surface area, thereby enhancing the interaction between the asphalt binder and the biochar. This property can enhance the performance of the asphalt binder by creating a skeletal structure between the biochar and the asphalt binder [31,61–64].

## 4.2. Conventional asphalt binder test results

Fig 5 presents the penetration test results, which indicate that the penetration value of the asphalt binder decreased with increasing contents of all biochar additives. It is seen that the penetration value of the neat asphalt binder, which is 51.2 dmm, is 43.9 dmm with 5% biochar content obtained at 300 °C temperature and rapid cooling condition. At 10% and 15% biochar content, penetration values were determined as 36.4 and 31.6 dmm, respectively. Similar changes were observed in the penetration values of biochar-modified asphalt binders obtained at 450 and 600 °C. As can be seen in Fig 5, the smallest penetration value of 30.6 dmm was reached at 15% asphalt binder content of biochar cooled rapidly at 450 °C (450R15).

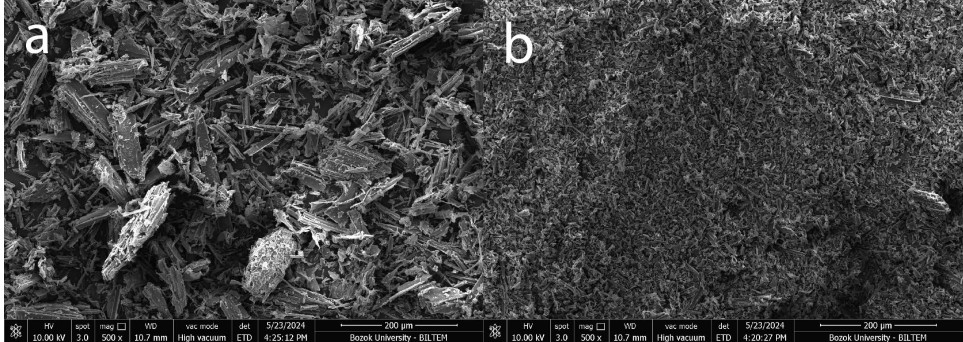

**Fig 4. SEM images of biochars (a: slow cooling condition – b: rapid cooling condition).**

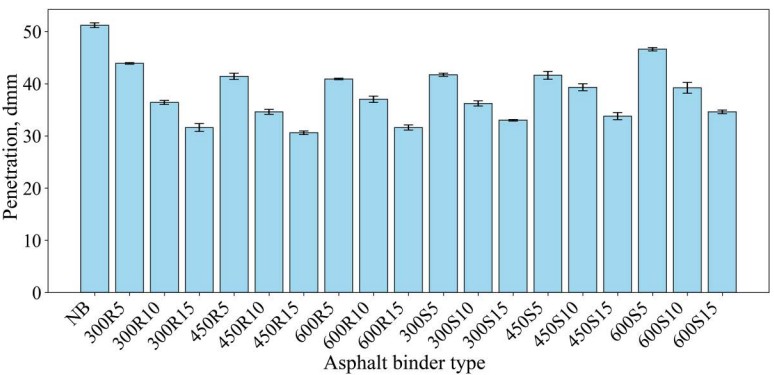

**Fig 5. Penetration values.**

As shown in Fig 6, the addition of biochar increases the softening point of asphalt binders as the biochar content in the asphalt binder increases.

PI values of asphalt binders are given in Fig 7. The PI values of neat and biochar-modified asphalt binders were compared, with PI being an indicator of temperature susceptibility. The PI value of neat asphalt binder was found to be −1.66, and it was generally observed that PI values increased with the addition of biochar. The highest PI values were obtained from asphalt binders modified with 15% biochars obtained by rapid cooling at 300 and 450 ˚C. This indicates that the addition of biochar, especially obtained under rapid cooling condition, can reduce the temperature susceptibility of asphalt binders and improve their high temperature performance. Due to its inherently porous and fibrous structure [63], biochar strengthens the asphalt binder, contributing to improved thermal stability.

## 4.3. RV test results

The rotational viscosity values of both neat and modified asphalt binders have been measured at intervals of 10°C within the range of 130–170°C. The outcomes of this test are illustrated in Table 4. In all groups of biochar-modified asphalt

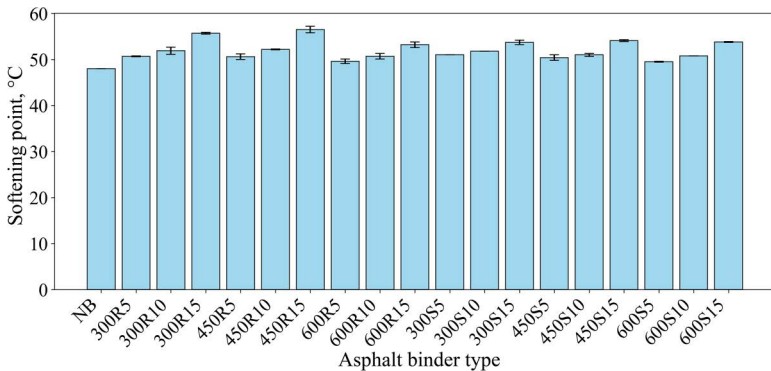

**Fig 6. Softening point values.**

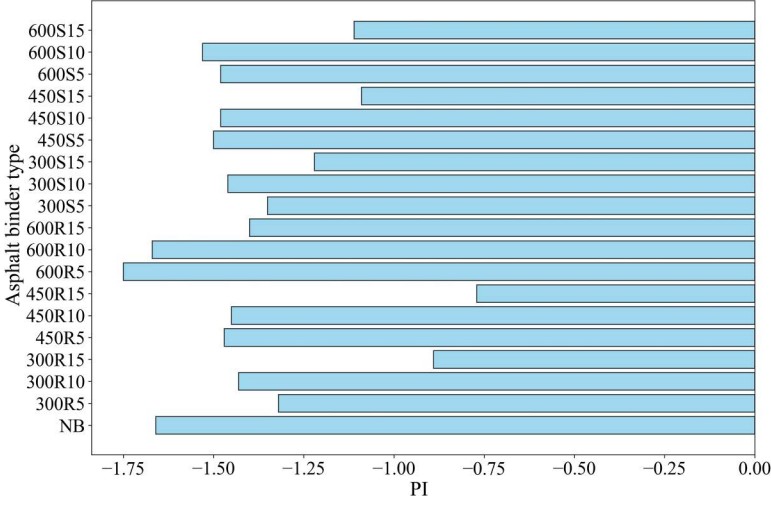

**Fig 7. PI values of asphalt binders.**

binders, an increase in biochar content has resulted in increased viscosity values of the asphalt binders. It has been established that the biochar ratio is most effective at temperatures up to 150°C; beyond this threshold, increases in the biochar ratio provide comparable viscosity values. The incorporation of biochar has been found to enhance the hardening of the asphalt binder, resulting in increased viscosity values, with the modified asphalt binder groups 300R15 and 450R15 exhibiting the most significant hardening effects. As can be seen from Table 4 the viscosity increases with each additional modifier dose.

### 4.4. DSR test results of unaged asphalt binders

DSR test results of unaged asphalt binders are given in Table 5. According to the AASHTO M320 standard, G*/sinδ values of unaged asphalt binders are required to be greater than 1.0 kPa. For neat asphalt binder, the highest temperature at which this condition is met was determined as 64 °C. The experimental results showed that all biochar groups increased the high temperature grade of the neat asphalt binder to 70 or 76 °C. It was determined that the asphalt binder in which the high temperature grade was achieved at the highest temperature of 76 °C were 300R15, 450R15, 300S15 and 450S15 type asphalt binders. Considering this situation, it can be concluded that the biochars obtained at pyrolysis temperatures of 300 and 450 °C are more effective, as the biochars derived from industrial hemp stalks were found to enhance the permanent deformation resistance of the asphalt binder. It is also noteworthy that at 70 and 76 °C, the rutting parameters of biochar modified asphalt binders obtained by rapid cooling were higher than those of biochar-modified asphalt binders obtained by slow cooling. For example, at 76 °C, the rutting parameter was determined to be 1.2 kPa for 300R15 and 450R15 types of asphalt binders, while it was 1.0 kPa for 300S15 and 450S15 types of asphalt binders.

**Table 4. Rotational viscosity values of the asphalt binders.**

| Asphalt binder type | Temperature, °C | | | | |
|---|---|---|---|---|---|
| | 130 | 140 | 150 | 160 | 170 |
| | Viscosity, Pa.s | | | | |
| NB | 0.58 | 0.34 | 0.20 | 0.13 | 0.08 |
| 300R5 | 0.79 | 0.46 | 0.28 | 0.18 | 0.11 |
| 300R10 | 1.26 | 0.74 | 0.45 | 0.29 | 0.19 |
| 300R15 | 2.58 | 1.51 | 0.95 | 0.61 | 0.41 |
| 450R5 | 0.80 | 0.46 | 0.29 | 0.19 | 0.11 |
| 450R10 | 1.26 | 0.74 | 0.45 | 0.29 | 0.19 |
| 450R15 | 2.63 | 1.54 | 0.95 | 0.61 | 0.41 |
| 600R5 | 0.80 | 0.46 | 0.28 | 0.18 | 0.11 |
| 600R10 | 1.05 | 0.61 | 0.38 | 0.24 | 0.15 |
| 600R15 | 1.70 | 0.99 | 0.61 | 0.39 | 0.25 |
| 300S5 | 0.78 | 0.45 | 0.28 | 0.16 | 0.10 |
| 300S10 | 1.11 | 0.65 | 0.41 | 0.25 | 0.16 |
| 300S15 | 1.99 | 1.18 | 0.73 | 0.46 | 0.30 |
| 450S5 | 0.78 | 0.45 | 0.28 | 0.16 | 0.10 |
| 450S10 | 1.10 | 0.64 | 0.40 | 0.25 | 0.16 |
| 450S15 | 1.88 | 1.10 | 0.68 | 0.44 | 0.29 |
| 600S5 | 0.73 | 0.43 | 0.26 | 0.16 | 0.10 |
| 600S10 | 1.01 | 0.59 | 0.35 | 0.23 | 0.15 |
| 600S15 | 1.55 | 0.90 | 0.55 | 0.35 | 0.24 |

**4.4.1. Statistical analysis of DSR test results.** Statistical analyses were conducted to evaluate the influence of various factors on the rutting parameter (G*/sinδ) at a 0.05 significance level (95% confidence level) using the Statistical Package for Social Sciences (SPSS, 2003). The analysis of variance (ANOVA) was performed using the general linear model (GLM) procedure. The independent and dependent variables considered in the analysis are presented in Table 6. Since the cooling condition one of the independent variables addressed is qualitative variable, "level set" was assigned to this variable to observe its contribution to the model. The variables constituting the level sets are referred to as dummy

**Table 5. Unaged asphalt binder DSR test results.**

| Asphalt binder type | G*/sinδ, kPa | | | | Test temperature that satisfies the condition, °C (G*/sinδ≥1.0 kPa) |
|---|---|---|---|---|---|
| | **58 °C** | **64 °C** | **70 °C** | **76 °C** | |
| NB | 3.9 | **1.8** | 0.9 | – | 64 |
| 300R5 | 5.0 | 2.3 | **1.1** | – | 70 |
| 300R10 | 6.4 | 3.0 | **1.5** | 0.8 | 70 |
| 300R15 | 9.4 | 4.5 | **2.2** | **1.2** | 76 |
| 450R5 | 5.2 | 2.5 | **1.2** | 0.6 | 70 |
| 450R10 | 7.1 | 3.1 | **1.5** | 0.8 | 70 |
| 450R15 | 9.8 | 4.7 | 2.1 | **1.2** | 76 |
| 600R5 | 5.7 | 2.6 | **1.3** | 0.7 | 70 |
| 600R10 | 6.7 | 3.1 | **1.5** | 0.8 | 70 |
| 600R15 | 8.2 | 3.7 | **1.8** | 0.9 | 70 |
| 300S5 | 4.7 | 2.2 | **1.0** | – | 70 |
| 300S10 | 6.1 | 2.8 | **1.4** | 0.7 | 70 |
| 300S15 | 9.0 | 3.9 | 1.9 | **1.0** | 76 |
| 450S5 | 4.9 | 2.2 | **1.0** | – | 70 |
| 450S10 | 5.9 | 2.6 | **1.2** | – | 70 |
| 450S15 | 9.0 | 4.0 | 1.9 | **1.0** | 76 |
| 600S5 | 4.8 | 2.2 | **1.0** | – | 70 |
| 600S10 | 6.0 | 2.8 | **1.4** | 0.7 | 70 |
| 600S15 | 8.1 | 3.6 | **1.7** | 0.9 | 70 |

**Table 6. Variables used in variance analysis.**

| Independent variables | Level | Definition |
|---|---|---|
| Production Temperature (PTEMP) | 300 °C<br>450 °C<br>600 °C | Biochar production temperature |
| Cooling Condition (COOLING) | Rapid (1)<br>Slow (2) | Biochar cooling conditions |
| Biochar Utilisation Content (CONT) | 0<br>%5<br>%10<br>%15 | Used biochar contents |
| DSR Test Temperature (TESTTEM) | 58 °C<br>64 °C<br>70 °C | DSR test temperatures |
| **Dependent variables** | **Definition** | |
| G*/sinδ | Rutting parameter | |

 

variables [65]. Dummy variables are shown in parentheses in Table 6. Numerically, dummy variables have no significance; they only indicate the levels of the independent variables considered [65].

The rutting resistance parameter model based on the DSR test results is given below.

$$G^*/\sin\delta \; = \; \mu \; + \; \alpha_1\,\text{PTEMP} \; + \; \alpha_2\text{COOLING} \; + \; \alpha_3\text{CONT} \; + \; \alpha_4\text{TESTTEM}$$
$$+ \; \alpha_5\text{PTEMP*CONT} \; + \; \alpha_6\text{COOLING*CONT}$$
$$+ \; \alpha_7\text{COOLING*TESTTEM} \; + \; \alpha_8\text{CONT*TESTTEM} \tag{2}$$

In the model;

μ: Constant coefficient.

$\alpha_i$: Multiple regression coefficient.

Details about the rutting parameter model established using the DSR test results is provided in Table 7.

The results presented in Table 7 indicate that the biochar production temperature, cooling method, biochar content in the asphalt binder, and the DSR test temperature have a significant impact on the rutting resistance parameter. Consequently, these factors should be carefully considered when modifying asphalt binders with biochar. The interactions between production temperature and cooling, as well as production temperature and DSR test temperature, were found to have no effect on the rutting resistance parameter. The relationship between the rutting resistance parameters determined experimentally and those predicted by the model is illustrated in Fig 8. As shown, the correlation between the observed and predicted values is exceptionally strong, with a coefficient of determination of $R^2 = 0.997$. This high value indicates that the developed model can be used for the prediction of the rutting resistance parameter.

The Duncan's multiple range test, a type of multiple comparison test, was also utilised in the analysis of variance. This test determined the significance of differences in rutting resistance parameters across various levels of each independent variable (e.g., production temperature: 300, 450, and 600 °C). In Duncan's multiple range test, if the differences in rutting resistance parameters for each level of the independent variables are significant ($p < 0.05$), different letters are assigned to the rutting resistance parameters; if the differences are insignificant ($p > 0.05$), the same letter is assigned to the rutting resistance parameters. The rutting resistance values for the levels of each independent variable are given in Table 8.

**Table 7. Statistical model details.**

| Statistical parameter | Values | Definition |
|---|---|---|
| $R^2$ | 0.997 | $(P_r > F) < 0,0001$ (significant) |
| RMSE, kPa | 0.0171 | |
| **Parameter/Interaction** | **$P_r > F$** | **Definition** |
| PTEMP | 0.0122 | S |
| COOLING | <0.0001 | S |
| CONT | <0.0001 | S |
| TESTTEMP | <0.0001 | S |
| PTEMP*CONT | <0.0001 | S |
| COOLING*CONT | 0.0024 | S |
| COOLING*TESTTEMP | 0.0265 | S |
| CONT*TESTTEMP | <0.0001 | S |

In the Table 7, $R^2$: Coefficient of determination; RMSE: Root mean square errors; S: Significant; NS: Not significant.

Table 8 illustrates that, according to Duncan's multiple range test, the rutting resistance parameters were determined as 3.53 kPa for the biochar-modified asphalt binder produced by rapid cooling and 3.24 kPa for the asphalt binder produced by slow cooling. From Table 8, it can be seen that adding biochar produced by rapid cooling to the asphalt binder increases the rutting resistance parameter more than slow cooling. This indicates that pavements made with biochar-modified asphalt binder, produced by rapid cooling, may exhibit better rutting resistance. The content of biochars added to the asphalt binder causes a significant difference on the rutting resistance parameter of the asphalt binders modified with biochars and the rutting resistance parameter increases as the content of biochars added increases. The rutting resistance parameter was found to be 2.17, 2.83, 3.56 and 4.97 kPa for 0%, 5%, 10% and 15% biochars, respectively.

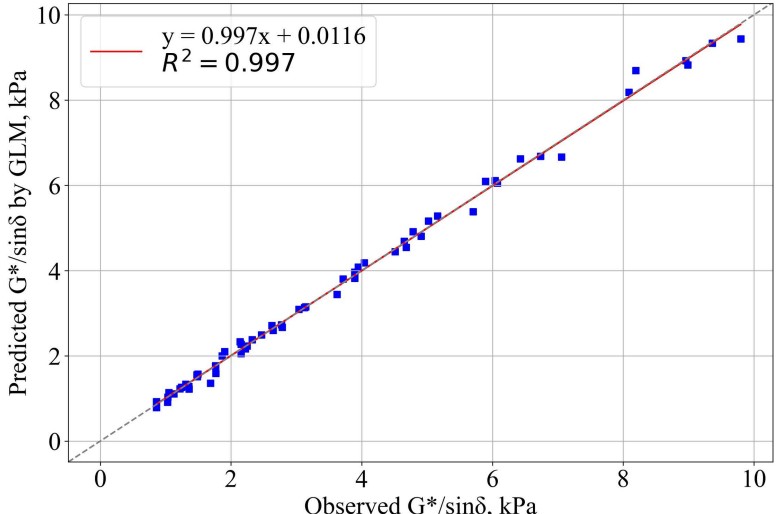

**Fig 8. The relationship between G\*/sinδ values determined experimentally and G\*/sinδ values predicted by the model.**

**Table 8. Duncan's multiple range test results.**

| Independent values levels | Rutting parameter, kPa |
|---|---|
| **Biochar production temperature** | |
| 300 °C | 3.39a,b |
| 450 °C | 3.46a |
| 600 °C | 3.30b |
| **Cooling condition** | |
| Rapid | 3.53a |
| Slow | 3.24b |
| **Biochar content** | |
| 0 | 2.17d |
| 5 | 2.83c |
| 10 | 3.56b |
| 15 | 4.97a |
| **DSR test temperature** | |
| 58 °C | 6.05a |
| 64 °C | 2.77b |
| 70 °C | 1.33c |

The DSR test results conducted at three different temperatures on both neat asphalt binders and biochar-modified asphalt binders indicated that the test temperature significantly influenced the rutting resistance parameter. As expected, the rutting resistance parameter decreased with the increase in test temperature. The rutting parameters at production temperatures of 300, 450, and 600 °C were 3.39, 3.46, and 3.30 kPa respectively. The differences between production temperatures of 300 °C and 450 °C, as well as between 300 °C and 600 °C, were considered insignificant (0.07 and 0.09 kPa, respectively); however, the difference between 450 °C and 600 °C was significant (0.16 kPa). The difference in the rutting resistance parameters at the production temperatures of 300 °C and 450 °C is negligible, therefore, it would be appropriate to choose 300 °C since biochar production at 300 °C is more economical in terms of energy consumption.

For these reasons, it was decided to conduct short- and long-term aging processes, followed by performance tests, on modified asphalt binders incorporating biochars produced exclusively at a pyrolysis temperature of 300 °C using by rapid cooling condition.

## 4.5. DSR test results of RTFOT aged asphalt binders

Table 9 shows the comparison of rutting parameters (G*/sinδ) of RTFOT aged asphalt binders. According to the test results, it is seen that the specification limit of 2.2 kPa for the G*/sinδ values of RTFOT aged asphalt binders was achieved at 76 °C for 300R15 asphalt binder and at 70 °C for other biochar modified asphalt binders. At all DSR test temperatures, the G*/sinδ values of 5% biochar modified asphalt binders were found to be lower than those of neat asphalt binders. This is due to the higher resistance of 5% biochar modified asphalt binders to aging (indicated in section 4.5.1). When more than 5% biochar is used, the stiffness of the asphalt binder increases with the biochar content increment, thereby enhancing its resistance to permanent deformations.

### 4.5.1. Evaluation of aging performance of binders.
Complex modulus aging indices (CAI), phase angle aging indices (PAI) and rheological aging indices (RAI) of the asphalt binders are presented in Fig 9.

A larger CAI value indicates that the asphalt binder is more susceptible to aging [66]. Fig 9a shows that asphalt binders containing biochar additives have lower CAI values and these asphalt binders have lower susceptibility to aging compared to neat asphalt binder. At 58 °C, the CAI value of the neat asphalt binder was found to be 3.38, while this value was found to be 2.14 for 300R5 asphalt binder. In this case, it was determined that the biochar addition increased the aging resistance of the asphalt binder by 36.7%.

The PAI values in Fig 9b increased with the addition of biochar. The increase in PAI values indicates that the aging performance of asphalt binders has improved [54]. In this case, it is concluded that modification of asphalt binder with biochar improves the aging performance of asphalt binder. It is seen that biochar ratio and biochar cooling type have no significant effect on PAI values.

**Table 9. DSR test results of RTFOT aged asphalt binders.**

| Asphalt binder type | G*/sinδ, kPa (RTFOT aged) | | | |
|---|---|---|---|---|
| | 58 °C | 64 °C | 70 °C | 76 °C |
| NB | 13.4 | 5.6 | 2.8 | – |
| 300R5 | 10.8 | 4.9 | 2.2 | – |
| 300R10 | 15.9 | 7.2 | 3.2 | – |
| 300R15 | 23.0 | 9.7 | 4.5 | **2.2** |
| 300S5 | 11.3 | 5 | 2.3 | – |
| 300S10 | 13.6 | 5.9 | 2.9 | – |
| 300S15 | 20.3 | 10.1 | 4.2 | – |

RAI values were determined based on the rutting parameter. Accordingly, it was determined that biochar additive tends to decrease the RAI values of asphalt binders. The lower RAI value indicates that the asphalt binder has lower aging susceptibility [53,67]. In this case, it is seen that biochar additive decreases the aging susceptibility of the asphalt binder. According to the RAI values in Fig 9c, the asphalt binders with the highest aging performance were determined as 300R5, 300S10 and 300R5 at 58, 64 and 70 °C, respectively. Thus, it is seen that biochar addition can increase the aging resistance of neat asphalt binder by approximately 35%.

In particular, it was determined that asphalt binders modified with 5% biochar, produced at a pyrolysis temperature of 300°C and subjected to a rapid cooling process, exhibited higher aging resistance.

### 4.6. Multiple Stress Creep Recovery (MSCR) test results

MSCR test was performed on RTFO aged neat and biochar modified asphalt binders at 64 °C temperature at 0.1 and 3.2 kPa stress levels. The Jnr and R values obtained as a result of the test are presented in Fig 10. Regarding the Jnr parameter, the lower its value, the higher the resistance of the asphalt binder to rutting due to a lower residual deformation after a creep cycle [36,68]. When Jnr values at 0.1 stress levels were analysed, it was observed that 300R10 and 300R15 asphalt binders provided 20.7% and 48.6% reduction compared to neat asphalt binder, respectively. In 300S15 asphalt binder, a 35% reduction was observed compared to neat asphalt binder. The asphalt binder with the highest rutting resistance was determined as 300R15. At 3.2 kPa stress level, Jnr values of all asphalt binders increased compared to 0.1

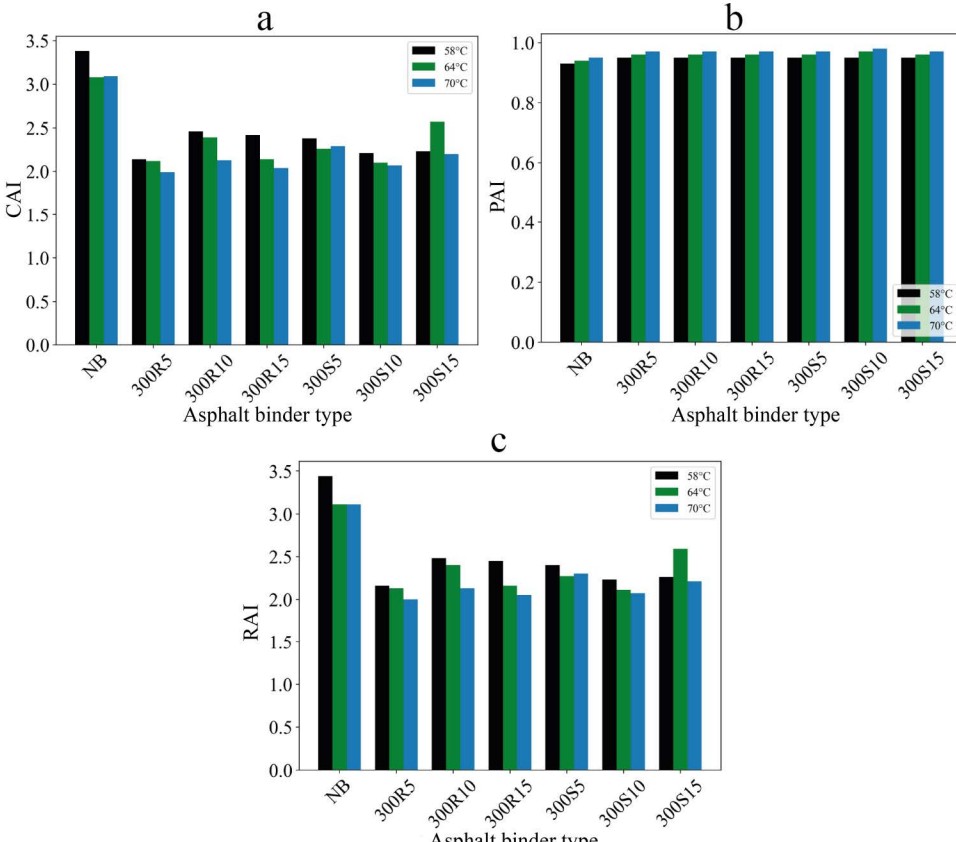

**Fig 9. (a) CAI, (b) PAI and (c) RAI aging indices.**

stress level. The corresponding traffic class for each asphalt binder type is illustrated in Fig 10b. At 3.2 kPa stress level, the asphalt binder with the lowest Jnr is 300R15. According to MSCR classification of neat asphalt binder, it is classified at H (Heavy Traffic Grade) level, while 300R15 asphalt binder is classified at V (Very Heavy Traffic Grade) level [69]. The recovery percentage results in the 0.1 kPa stress level showed that asphalt binders of types 300R10, 300R15, 300S10 and 300S15 exhibited more elastic behaviour than neat asphalt binder. These asphalt binders with high recovery rate allow for more recovery of deformations.

The difference in ($Jnr_{diff}$) between 0.1 and 3.2 kPa stress levels is presented in Fig 11. The difference between the creep recovery ($Jnr_{diff}$) of the asphalt binders at 0.1 and 3.2 kPa stress levels is greater than 75%, indicating susceptibility to rutting [70]. It is seen that the $Jnr_{diff}$ values of all asphalt binders at 64 °C are below 75%, and the lowest value of 14% belongs to asphalt binder 300R5. The low $Jnr_{diff}$ value of the 300R5 type asphalt binder may be due to its high resistance to aging.

### 4.7. Bending Beam Rheometer (BBR) test results

In order to evaluate the low temperature performance of asphalt binders, BBR tests were applied to PAV aged asphalt binders. Table 10 shows the creep stiffness and m-value values of asphalt binders at −6 and −12 °C temperatures and

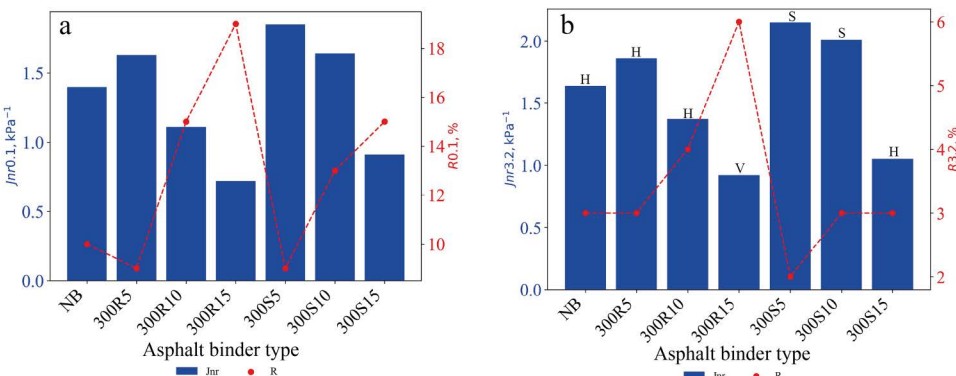

**Fig 10. (a) Jnr and R values at 0.1 kPa and (b) Jnr and R values at 3.2 kPa, and traffic class.**

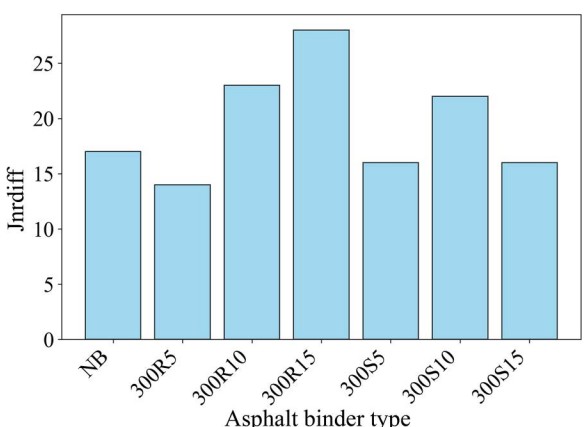

**Fig 11. $Jnr_{diff}$ values.**

low temperature performance grade. It was determined that biochar addition increased the creep stiffness of neat asphalt binder at both temperatures. Table 10 shows that the low temperature performance grades of asphalt binders modified with biochar have decreased by one grade. The results are consistent with the literature and it was determined that the use of biochar can make asphalt binders more brittle at low temperatures [34,53,62]. In the BBR test carried out at −12 °C, it is noteworthy that while the modified asphalt binder containing 5% biochar met the stiffness requirement, it failed to meet the m value requirement by a very small margin.

## 4.8. DSR test results of PAV aged asphalt binders

While the high temperature performance grades of the asphalt binders are determined with the help of Tables 5 and 9, the low temperature performance grades are given in Table 10 and the DSR test temperatures to be carried out to determine the fatigue resistance of asphalt binders with the help of high and low temperatures are calculated. The temperatures at which the DSR test will be carried out and the test results are given in Table 11. The value of G*sinδ represents the stiffness at moderate temperatures and serves as an indicator of fatigue resistance. A decrease in G*sinδ indicates an increase in fatigue resistance, while an increase in G*sinδ suggests a decrease in fatigue resistance. As shown in Table 11, at 25°C, it is observed that for all types of asphalt binders modified with biochar, the G*sinδ value increases compared to the neat asphalt binder, indicating a decrease in fatigue resistance. However, at 28°C and 31°C, the G*sinδ value decreases for the 300S5 and 300S10 asphalt binder types compared to the neat asphalt binder, indicating an increase in fatigue resistance, while the opposite trend is observed for the other biochar modified asphalt binders. From these results, it can be concluded that asphalt binders modified with 5–10% biochar obtained through slow cooling exhibit higher fatigue resistance performance compared to those modified with biochar obtained through rapid cooling and the neat binders.

Table 10. BBR test results.

| Asphalt binder type | Creep stiffness (S), MPa | | m-value | | Low Temperature PG |
|---|---|---|---|---|---|
| | −6°C | −12°C | −6°C | −12°C | |
| NB | 64.77 | 203.62 | 0.38 | 0.30 | −22 |
| 300R5 | 98.57 | 232.54 | 0.34 | 0.26 | −16 |
| 300R10 | 143.73 | 316.79 | 0.32 | 0.24 | −16 |
| 300R15 | 180.42 | 378.93 | 0.32 | 0.24 | −16 |
| 300S5 | 111.71 | 265.73 | 0.35 | 0.27 | −16 |
| 300S10 | 127.54 | 318.54 | 0.33 | 0.26 | −16 |
| 300S15 | 176.06 | 390.83 | 0.32 | 0.25 | −16 |

Table 11. DSR test results of PAV aged asphalt binders.

| Asphalt binder type | High temperature PG | Low temperature PG | Indermediate test temperature, ˚C [(max+min)/2]+4 | G*.sinδ, kPa, (PAV-aged) | | |
|---|---|---|---|---|---|---|
| | | | | 25 °C | 28 °C | 31 °C |
| NB | 64 | −22 | 25 | 2776 | 2403 | 1528 |
| 300R5 | 70 | −16 | 31 | 3797 | 2696 | 1828 |
| 300R10 | 70 | −16 | 31 | 4405 | 2972 | 2003 |
| 300R15 | 76 | −16 | 34 | 3351 | 2471 | 1776 |
| 300S5 | 70 | −16 | 31 | 2958 | 2051 | 1376 |
| 300S10 | 70 | −16 | 31 | 3368 | 2237 | 1398 |
| 300S15 | 70 | −16 | 31 | 4673 | 3137 | 2089 |

In accordance with the Superpave specifications, the maximum allowable G*sinδ value for PAV-aged asphalt binders is 5000 kPa at intermediate test temperatures. The G*sinδ values for both neat and biochar modified asphalt binders were found to be below this threshold at their intermediate test temperatures, thereby meeting the Superpave requirements.

## 5. Conclusion

In this study, the usability of biochar derived from industrial hemp stalks under varying pyrolysis conditions as an additive for asphalt binders was evaluated through conventional and rheological asphalt binder tests as well as various material characterization techniques. Based on the findings, the following conclusions were made.

Adding biochar to the asphalt binders demonstrates a stiffening effect, as indicated by empirical parameters such as the softening point and penetration at 25°C, as well as the rutting parameter (G*/sin δ) determined from DSR results.

The rotational viscosity values of the asphalt binder increase proportionally with the amount of biochar used, implying a slight increase in the mixing and compaction temperatures of hot mix asphalt.

SEM analysis showed that biochars produced using the rapid cooling method may have a high specific surface area, which improves the interaction between the asphalt binder and the biochar. This property can enhance the performance of the asphalt binder by forming a skeletal structure between the biochar and the asphalt binder.

Statistical analysis revealed that the biochar production temperature, cooling method, biochar content in the asphalt binder, and DSR test temperature significantly influence the rutting resistance parameter. According to Duncan's multiple range test, it is found that adding biochar produced by rapid cooling to the asphalt binder increases the rutting resistance parameter more than slow cooling. No significant difference was observed in terms of rutting parameters of asphalt binders between the pyrolysis temperatures used in biochar production. It was determined that the lower pyrolysis temperature of 300 °C was more appropriate in terms of energy efficiency. So, the optimal production parameters for biochar were identified as a temperature of 300 °C combined with rapid cooling.

The complex modulus aging index (CAI), phase angle aging index (PAI) and rheological aging index (RAI) determined using G* and δ values of unaged and RTFO aged asphalt binders showed that modification of asphalt binder with biochar improved the aging performance of asphalt binder. It was determined that asphalt binders modified by using 5% of biochar produced at 300 °C pyrolysis temperature and rapid cooling process had higher aging resistance.

It was determined that the low temperature performance grades of asphalt binders modified with biochar have decreased by one grade. In the BBR test carried out at −12 °C, it is noteworthy that while the modified asphalt binder containing 5% biochar met the stiffness requirement, it failed to meet the m value requirement by a very small margin. The biochar additive had a negative effect on the low-temperature performance of the asphalt binder.

In summary, it was found that the rutting resistance of asphalt binders modified with industrial hemp stalk biochar was enhanced. The optimal conditions for biochar production, intended for this purpose, were identified as a pyrolysis temperature of 300°C coupled with rapid cooling. Although the incorporation of hemp stalk biochar may reduce the low-temperature performance of asphalt, this effect is less pronounced in hot climate regions, where the low-temperature performance requirements are less stringent, making hemp stalk biochar a viable alternative in the pavement industry in such regions.

## Supporting information

**S1 File.  Minimal data set.**
(DOCX)

## Author contributions

**Conceptualization:** İbrahim Aslan, Yuksel Tasdemir.

**Formal analysis:** Funda Tasdemir.

**Funding acquisition:** İbrahim Aslan, Yuksel Tasdemir.

**Investigation:** İbrahim Aslan, Yuksel Tasdemir.

**Methodology:** İbrahim Aslan, Funda Tasdemir, Yuksel Tasdemir.

**Project administration:** Yuksel Tasdemir.

**Resources:** Yuksel Tasdemir.

**Software:** Funda Tasdemir.

**Supervision:** Yuksel Tasdemir.

**Visualization:** İbrahim Aslan.

**Writing – original draft:** İbrahim Aslan, Funda Tasdemir.

**Writing – review & editing:** Funda Tasdemir, Yuksel Tasdemir.

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
