## [Decision Letter · Decision Letter 0]

PONE-D-25-02281Utilization of Biochar Derived from Industrial Hemp Stalks for the Modification of Bituminous BindersPLOS ONE

Dear Dr. Aslan,

Thank you for submitting your manuscript to PLOS ONE. After careful consideration, we feel that it has merit but does not fully meet PLOS ONE’s publication criteria as it currently stands. Therefore, we invite you to submit a revised version of the manuscript that addresses the points raised during the review process.

We look forward to receiving your revised manuscript.

Kind regards,

Mayank Sukhija

Academic Editor

PLOS ONE

Journal Requirements:

“This work was supported by Yozgat Bozok University with project code FKA-2022-1005 and the Scientific and Technological Research Council of Türkiye (TUBITAK) with project code 122M886.”

**Additional Editor Comments:**

I have received the comments from the Reviewers. You can see they are recommending significant revisions. Kindly incorporate the comments. It is suggested to do Life cycle assessment for proving the sustainability aspect of Biochar.

Reviewers' comments:

Reviewer's Responses to Questions

**Comments to the Author**

1. Is the manuscript technically sound, and do the data support the conclusions?

Reviewer #1: Yes

Reviewer #2: Yes

Reviewer #3: Yes

2. Has the statistical analysis been performed appropriately and rigorously? 

Reviewer #1: Yes

Reviewer #2: Yes

Reviewer #3: Yes

3. Have the authors made all data underlying the findings in their manuscript fully available?

Reviewer #1: Yes

Reviewer #2: Yes

Reviewer #3: No

4. Is the manuscript presented in an intelligible fashion and written in standard English?

Reviewer #1: Yes

Reviewer #2: Yes

Reviewer #3: Yes

5. Review Comments to the Author

Reviewer #1: The authors have worked on an interesting and relevant area. Although many studies have been conducted on biochar modified asphalt binders, the influence of pyrolysis conditions (rate of cooling, pyrolysis temperature, residence time, etc) on the properties of resulting modified binders are poorly understood. Some comments on the manuscript are as follows:

1. Introduction: This section should include previous studies on variation of pyrolysis conditions, and should also explain the significance of the pyrolysis parameters (temperature and rate of cooling). Authors are encouraged to check for papers from chemical / energy engineering domain to enhance the quality of the introduction.

2. In Introduction section, briefly add a paragraph to explain the need of the study and the novelty.

3. L74-77: Please do not include the conclusions of your study in the introduction section.

4. Para 2.2: Please add several pics to show the processing of the material used in pyrolysis, pyrolysis reactor (actual picture or a flow chart), process of slow and rapid cooling, and any other relevant picture.

5. L187-188: Please add the results of previous studies in Table 3 in support of your statement that “elemental analysis results…were found similar to those of previous studies.”

6. Para 4.1: In Para 3.1, you mention conducting CHNS analysis; however the results presented are for CHN.

7. Para 4.1: Please add some scientific insights into the variation of the results. What causes changes in CHN contents on slow / rapid cooling? What happens when the pyrolysis temperature is increased?

8. Table 4: The table is data intensive; suggest presenting them in the form of a graph.

9. Para 4.4.1: Why did you use ‘dummy’ variables for numerical data (pyrolysis temperature, biochar content, DSR temperature)? Dummy variable is justified for cooling condition as the levels are not numeric.

10. Revise your statistical model by removing the non-significant terms.

Reviewer #2: Utilization of Biochar Derived from Industrial Hemp Stalks for the Modification of Bituminous Binders

This paper focuses on determining the optimal production temperature and cooling conditions for pyrolysis in biochar production. A series of tests were done, and careful conclusions have been drawn. In its current state, the paper is overall insightful and informative. Below are a few comments.

1. Line 46: Add a reference

2. Line 103: Sentence seems to have a grammatical problem

3. Line 161: Add a reference for that statement

4. Under section 4.2, what do you think caused those results

Reviewer #3: The topic of the presented paper is interesting and a comprehensive experimental program was performed. However, the problem statement and the objectives of the article were not defined well. Also, the results did not indicate that the biochar derived in this study can be an appropriate additive that could be useful for bitumen modification. It is actually a filler and it is required to perform a LCA to show the benefits of using such an additive for bitumen modification. The following comments also must be properly addressed:

1. It is needed to revise the title of the article as it does not reflect the paper’s content well.

2. In the beginning of the abstract, it is mentioned “This study aimed to determine the optimal pyrolysis temperature and cooling conditions to produce biochar from industrial hemp stalks” but the authors did not state the optimal temperature and cooling conditions.

3. The abstract must be written quantitatively to reflect the effect of adding biochar to bitumen by using increasing and decreasing values. For example, the authors stated that “It was determined that the biochar additive had a minimal negative effect on the low-temperature performance of the binder”. It is required to state a specific value for “minimal negative effect”.

4. As the main research gap, in the last paragraph of the introduction section, the authors stated “The primary focus of this study, unlike previous ones, is to determine the optimal production temperature and cooling conditions for pyrolysis in biochar production.” While it cannot be an appropriate research gap, it is not aligned with the literature review that the authors presented in previous paragraphs. There are lots of studies that focus on determining optimal pyrolysis temperature and cooling conditions. It is necessary to specify the main research gap that motivated the authors to conduct this study.

5. It is required that the authors expand their literature review to avoid claiming some statement like this “no prior study in the literature has utilized industrial hemp for the production of biochar.” For example, take a look at the following references:

Du, J., Zhang, F., Hu, J., Yang, S., Liu, H., & Wang, H. (2023). Co-pyrolysis of industrial hemp stems and waste plastics into biochar-based briquette: Product characteristics and reaction mechanisms. Fuel Processing Technology, 247, 107793.

Voglar, J., Prašnikar, A., Moser, K., Carlon, E., Schwabl, M., & Likozar, B. (2024). Pyrolysis of industrial hemp biomass from contaminated soil phytoremediation: kinetics, modelling transport phenomena and biochar-based metal reduction. Thermochimica acta, 742, 179899.

6. The results should not be mentioned in the introduction section as the authors stated “The optimal production parameters of biochar were determined as a temperature of 300 °C combined with rapid cooling”.

7. The authors must state the objectives of the study at the of the introduction section.

8. Replace “bituminous binder” with “bitumen” or “asphalt binder” in the whole manuscript.

9. Use SI unit of Pa.s or mPa.s instead of cP in Table 1.

10. It is needed to precisely define fast and slow cooling condition.

11. Why did the authors select three pyrolysis temperatures of 300, 450, and 600 degree of centigrade? Clarify it.

12. How did the authors select the biochar dosage for adding to bitumen? Clarify it.

13. It is required to specify the test temperature in which MSCR and BBR tests were performed.

14. In Figure 8, it is needed to specify the traffic level (S, H, V, or E) of each sample. It will help the readers to better understand the difference between samples.

15. Why did the authors consider two low-temperatures of -6 and -12 C? It is needed to measure flexural creep stiffness and m-value at lower temperatures like -18 and -24 C to better realize the effect of adding biochar to bitumen on its properties.

16. without a LCA, how did the authors claim “The findings indicate that biochar derived from industrial hemp stalks functions as a renewable, sustainable, and eco-friendly additive for bituminous binders” in the conclusion.

17. The whole manuscript needs a grammatical check.

18. The reference must be cited immediately after the authors’ names. For example, in line 56, it should be Dong et al. [11].

6. PLOS authors have the option to publish the peer review history of their article (what does this mean? ). If published, this will include your full peer review and any attached files.

**Do you want your identity to be public for this peer review?** For information about this choice, including consent withdrawal, please see our Privacy Policy .

Reviewer #1: No

Reviewer #2: No

Reviewer #3: No

---

## [Author Response · Author response to Decision Letter 1]

25 Mar 2025

Explanation of the corrections that have been made by following the reviewers comments.

Many thanks for useful comments to our paper, and good advice for changes!

Reviewer #1

1. Introduction: This section should include previous studies on variation of pyrolysis conditions, and should also explain the significance of the pyrolysis parameters (temperature and rate of cooling). Authors are encouraged to check for papers from chemical / energy engineering domain to enhance the quality of the introduction.

To explain the importance of pyrolysis parameters, the section of the introduction dealing with pyrolysis is re-written as follows:

“…source or additive material. Pyrolysis, the most commonly used and extensively studied process for biochar production, involves the thermal decomposition of biomass at elevated temperatures in an oxygen-free environment [6–8]. During pyrolysis, the biomass undergoes thermal decomposition, releasing volatile compounds and leaving behind a solid residue known as biochar. Biochar can be produced from a wide range of biomass, including agricultural waste, forestry waste, and municipal and industrial organic waste. The yield and properties of the biochar depend on various factors such as the type of biomass, the heating rate, pyrolysis temperature, and the residence time [9]. There are three different pyrolysis processes: slow, fast and flash pyrolysis. Slow pyrolysis yields more biochar products at low temperature and low heating rate, while fast pyrolysis yields higher bio-oil product at high temperature such and high heating rate. In flash pyrolysis, the heating rate is higher than in fast pyrolysis and the processing time is quite short [10]. In slow pyrolysis, characterized by low reaction temperatures and heating rates, the biochar yield is maximized, resulting in the production of high-quality biochar [11, 12]. Most of the previous laboratory scale research on slow pyrolysis investigated the effect of type of biomass, pyrolysis temperature, heating rate, and residence time, as the main operating variables that affected the biochar yield and properties [8, 13-19]. Biochar derived from different…”

The following references have been added to manuscript.

6. Tan H, Lee CT, Ong PY, Wong KY, Bong CPC, Li C, et al. A Review On The Comparison Between Slow Pyrolysis And Fast Pyrolysis On The Quality Of Lignocellulosic And Lignin-Based Biochar. IOP Conf Ser Mater Sci Eng. 2021;1051: 012075. doi:10.1088/1757-899X/1051/1/012075

7. Ng HS, Kee PE, Yim HS, Chen PT, Wei YH, Chi-Wei Lan J. Recent advances on the sustainable approaches for conversion and reutilization of food wastes to valuable bioproducts. Bioresour Technol. 2020;302: 122889. doi:10.1016/j.biortech.2020.122889

9. Zhou L. A Review of Biomass-Derived Biochar and Its Potential in Asphalt Pavement Engineering. Materials Science- Poland. 2024;42: 81–99. doi:10.2478/msp-2024-0022

10. Lee SY, Sankaran R, Chew KW, Tan CH, Krishnamoorthy R, Chu D-T, et al. Waste to bioenergy: a review on the recent conversion technologies. Bmc Energy. 2019;1. doi:10.1186/s42500-019-0004-7

11. Onay O, Kockar OM. Slow, fast and flash pyrolysis of rapeseed. Renew Energy. 2003;28: 2417–2433. doi:10.1016/S0960-1481(03)00137-X

12. Maqsood T, Dai J, Zhang Y, Guang M, Li B. Pyrolysis of plastic species: A review of resources and products. J Anal Appl Pyrolysis. 2021;159. doi:10.1016/j.jaap.2021.105295

13. Premchand P, Demichelis F, Chiaramonti D, Bensaid S, Fino D. Biochar production from slow pyrolysis of biomass under CO2 atmosphere: A review on the effect of CO2 medium on biochar production, characterisation, and environmental applications. J Environ Chem Eng. 2023;11: 110009. doi:10.1016/j.jece.2023.110009

14. Leng L, Huang H. An overview of the effect of pyrolysis process parameters on biochar stability. Bioresour Technol. 2018;270: 627–642. doi:10.1016/j.biortech.2018.09.030

15. Tag AT, Duman G, Ucar S, Yanik J. Effects of feedstock type and pyrolysis temperature on potential applications of biochar. J Anal Appl Pyrolysis. 2016;120: 200–206. doi:10.1016/j.jaap.2016.05.006

16. Elnour AY, Alghyamah AA, Shaikh HM, Poulose AM, Al-Zahrani SM, Anis A, et al. Effect of Pyrolysis Temperature on Biochar Microstructural Evolution, Physicochemical Characteristics, and Its Influence on Biochar/Polypropylene Composites. Applied Sciences 2019, Vol 9, Page 1149. 2019;9: 1149. doi:10.3390/APP9061149

17. Tomczyk A, Sokołowska Z, Science PB-R in E, 2020 undefined. Biochar physicochemical properties: pyrolysis temperature and feedstock kind effects. Springer. 2020;19: 191–215. doi:10.1007/s11157-020-09523-3

18. Li C, Hayashi J ichiro, Sun Y, Zhang L, Zhang S, Wang S, et al. Impact of heating rates on the evolution of function groups of the biochar from lignin pyrolysis. J Anal Appl Pyrolysis. 2021;155. doi:10.1016/j.jaap.2021.105031

19. Zhao B, O’Connor D, Zhang J, Peng T, Shen Z, Tsang DCW, et al. Effect of pyrolysis temperature, heating rate, and residence time on rapeseed stem derived biochar. J Clean Prod. 2018;174: 977–987. doi:10.1016/j.jclepro.2017.11.013

2. In Introduction section, briefly add a paragraph to explain the need of the study and the novelty.

The last part of the Introduction has been revised as follows to emphasize the study's significance and originality:

Factors affecting the yield and properties of biochar have also been taken into account in previous studies on biochar-modified asphalt binders and their mixtures [8]. No detailed study has been conducted on the cooling of biochar obtained after pyrolysis. In some studies, biochar was extracted from the steel reactor only after it had cooled to room temperature following the pyrolysis process, a procedure called as slow cooling, whereas other studies did not specify the cooling method. In this study, after pyrolysis, the hot biochar was immediately removed from the steel reactor, rapidly cooled by immersing it in water at 10°C, and then dried. This process was termed rapid cooling. In the study, biochar obtained by both methods was used in the experiments. The aim of the study was to determine the optimum pyrolysis temperature and cooling conditions for producing biochar from industrial hemp stalks to enhance asphalt binder performance. Additionally, the study evaluated the potential of industrial hemp stalks as a viable alternative in the pavement sector by analyzing the temperature sensitivity and high- and low-temperature performance of the binder with biochar incorporation. Although there are studies on biochar production from industrial hemp stalk for various purposes [39-42], as far as the authors are aware, no detailed research has been conducted on the performance properties of the asphalt binder incorporating industrial hemp stalk biochar.

The following references have been added to manuscript.

39. Du J, Zhang F, Hu J, Yang S. Co-pyrolysis of industrial hemp stems and waste plastics into biochar-based briquette: Product characteristics and reaction mechanisms. Fuel Processing Technology. 2023;247: 107793. doi:10.1016/j.fuproc.2023.107793

40. Voglar J, Prašnikar A, Moser K, Carlon E, Schwabl M, Likozar B. Pyrolysis of industrial hemp biomass from contaminated soil phytoremediation: Kinetics, modelling transport phenomena and biochar-based metal reduction. Thermochim Acta. 2024;742: 179899. doi:https://doi.org/10.1016/j.tca.2024.179899

41. Marrot L, Candelier K, Valette · Jérémy, Lanvin · Charline, Horvat B, Legan · Lea, et al. Valorization of hemp stalk waste through thermochemical conversion for energy and electrical applications. Waste Biomass Valorization. 2022;13: 2267–2285. doi:10.1007/s12649-021-01640-6

42. Atoloye ID IA, Adesina IS, Sharma H, Subedi KI, Liang C-L, Shahbazi A, et al. Hemp biochar impacts on selected biological soil health indicators across different soil types and moisture cycles. PLoS One. 2022;17. doi:10.1371/journal.pone.0264620

Note: The pyrolysis methods include slow pyrolysis, fast pyrolysis, and flash pyrolysis. The rapid cooling process mentioned in our study is a distinct operation from the fast pyrolysis method and is performed after biochar production. Therefore, the terms "fast pyrolysis" and "rapid cooling" refer to different concepts and should not be conflated.

3. L74-77: Please do not include the conclusions of your study in the introduction section.

The conclusions of our study given in line 74-77 (in old version) have been removed from the introduction section. In the new version of the manuscript, the results of our study are not given in the introduction.

L74-77: The optimal production parameters of biochar were determined as a temperature of 300 °C combined with rapid cooling. The incorporation of biochar enhanced the aging resistance and high-temperature performance of bituminous binders, although it resulted in a slight reduction in low-temperature performance.

4. Para 2.2: Please add several pics to show the processing of the material used in pyrolysis, pyrolysis reactor (actual picture or a flow chart), process of slow and rapid cooling, and any other relevant picture.

With respect to the reviewer comment, new figure is added in the text to explain graphically the pyrolysis process. The following sentences in red have been added to section 2.3.

“…The pyrolysis system employed for biochar production illustrated in Fig 1. Allowing the biochar to cool to room temperature within the steel reactor after the pyrolysis process is referred to as slow cooling, while rapidly cooling the biochar by immersing it in cold water immediately after its removal from the steel reactor is referred to as rapid cooling. The processes of the rapid cooling and slow cooling are schematically shown in Fig 2. After…”

Fig 1. (a) Pyrolysis system and (b) schematic illustration.

Fig 2. (a) Rapid cooling process and (b) slow cooling process.

5. L187-188: Please add the results of previous studies in Table 3 in support of your statement that “elemental analysis results…were found similar to those of previous studies.”

Sentences in lines 187-188 (now line 215-220) were re-written with additional reference to include the results of previous studies and compared with the results found in our study,and the rewritten part is marked in red.

“Table 3 presents the elemental analysis results of biochars produced from industrial hemp stalks of varying sizes under different pyrolysis conditions. The elemental analysis results indicated that raw hemp stalk with a particle size smaller than 0.25 mm contained 42.16% carbon, 5.995% hydrogen, and 0.257% nitrogen, while raw hemp stalk with a particle size of 10-30 mm contained 45.63% carbon, 6.435% hydrogen, and 0.36% nitrogen. Our findings consistent with the data reported in the literature, emphasizing that the carbon content of raw hemp biomass ranges from 45% to 49%, hydrogen from 5.6% to 6.2%, and nitrogen from 0.3% to 1.3% [41, 59, 60].

The following references have been added to manuscript.

41. Marrot L, Candelier K, Valette · Jérémy, Lanvin · Charline, Horvat B, Legan · Lea, et al. Valorization of hemp stalk waste through thermochemical conversion for energy and electrical applications. Waste Biomass Valorization. 2022;13: 2267–2285. doi:10.1007/s12649-021-01640-6.

59. Prade T, Finell M, Svensson SE, Mattsson JE. Effect of harvest date on combustion related fuel properties of industrial hemp (Cannabis sativa L.). Fuel. 2012;102: 592–604. doi:10.1016/j.fuel.2012.05.045

6. Para 4.1: In Para 3.1, you mention conducting CHNS analysis; however the results presented are for CHN.

As a result of the elemental analysis, S was found to be zero, so it is not given in the table. The results found are consistent with previous results. Since S was found to be zero as a result of the elemental analysis, the term “CHN analysis” is used instead of “CHNS analysis” throughout the article.

Pannipa Chaowana, Warinya Hnoocham, Sumate Chaiprapat, Piyawan Yimlamai, Korawit Chitbanyong, Kapphapaphim Wanitpinyo, Tanapon Chaisan, Yupadee Paopun, Sawitree Pisutpiched, Somwang Khantayanuwong, Buapan Puangsin, Utilization of hemp stalk as a potential resource for bioenergy, Materials Science for Energy Technologies,Volume 7, 2024, Pages 19-28, https://doi.org/10.1016/j.mset.2023.07.001.

7. Para 4.1: Please add some scientific insights into the variation of the results. What causes changes in CHN contents on slow / rapid cooling? What happens when the pyrolysis temperature is increased?

To explain the variation of the results clearly, in Section “4.1. Elemental and SEM Analysis Results” the following with red colour was added:

“Table 3 shows that the increase in the carbon amount of biochar obtained from the pyrolysis of small-sized biomass is high. The carbon content of hemp stalk biochar with a particle size smaller than 0.25 mm was found to range from 69.73% to 95.93%, while that of biochar with a particle size of 10–30 mm ranged from 65.5% to 83.85%, depending on the pyrolysis temperature and the cooling conditions. Changing the pyrolysis temperature and cooling conditions did not lead to a meaningful change in the observed increase in carbon. Hydrogen content exhibits an inverse relationship with temperature, decreasing as pyrolysis progresses. For small particles, hydrogen content drops from 3.884% at 300°C to 1.62% at 600°C under rapid cooling conditions. This reduction is attributed to the volatilization of hydrogen-containing compounds and the progressive formation of aromatic structures, which lead to lower H/C ratios and increased thermal stability of the biochar. Similarly, nitrogen content decreases with increasing temperature as nitrogenous compounds volatilize into gaseous emissions such as NH₃ and NOx.”

8. Table 4: The table is data intensive; suggest presenting them in the form of a graph.

As seen in the figure below, since the number of data is large and it is difficult to read the graph, it was thought that it would be appropriate to give the data in a table and the data is given in the Table 4.

9. Para 4.4.1: Why did you use ‘dummy’ variables for numerical data (pyrolysis temperature, biochar content, DSR temperature)? Dummy variable is justified for cooling condition as the levels are not numeric.

We would like to thank to the reviewer for pointing out this mistake. During the statistical analysis, level sets were used only for the cooling condition, but level sets were also assigned to other independent variables by mistake in Table 6 and the error was corrected now. Table 6 was rearranged as follows. The following change marked with red color has been made in the manuscript:

Since the cooling condition one of the independent variables addressed is qualitative variable, "level set" was assigned to this variable to observe its contribution to the model.

Table 6. Variables used in variance analysis.

Independent variables Level Definition

Production Temperature

(PTEMP) 300 °C

450 °C

600 °C Biochar production temperature

Cooling Condition

(COOLING) Rapid (1)

Slow (2) Biochar cooling conditions

Biochar Utilisation Content (CONT) 0

%5

%10

%15 Used biochar contents

DSR Test Temperature

(TESTTEM) 58 °C

64 °C

70 °C DSR test temperatures

Dependent variables Definition

G*/sinδ Rutting parameter

10. Revise your statistical model by removing the non-significant terms.

Statistical model was revised by removing the non-significant terms. Table 7 and Equation 2 were rearranged as follows.

Table 7. Statistical model details.

Statistical parameter Values Definition

R2 0.997 (Pr>F) <0,0001 (significant)

RMSE, kPa 0.0171

Parameter/Interaction Pr>F Definition

PTEMP 0.0122 S

COOLING <0.0001 S

CONT <0.0001 S

TESTTEMP <0.0001 S

PTEMP*CONT <0.0001 S

COOLING*CONT 0.0024 S

COOLING*TESTTEMP 0.0265 S

CONT*TESTTEMP <0.0001 S

G*/sinδ = ��+ �1 PTEMP + �2COOLING + �3CONT + �4TESTTEM + �5PTEMP*CONT +

+ �6COOLING*CONT + �7COOLING*TESTTEM + �8CONT*TESTTEM

Reviewer #2

1. Line 46: Add a reference

The following references have been added after the sentence "The primary applications of biochars currently lie within the agricultural sector." on line 45.

References:

20. Allohverdi T, Mohanty AK, Roy P, Misra M. A review on current status of biochar uses in agriculture. Molecu

---

## [Decision Letter · Decision Letter 1]

PONE-D-25-02281R1Utilization of Biochar Derived from Industrial Hemp Stalks with Various Cooling Methods for Asphalt Binder ModificationPLOS ONE

Dear Dr. Aslan,

Thank you for submitting your manuscript to PLOS ONE. After careful consideration, we feel that it has merit but does not fully meet PLOS ONE’s publication criteria as it currently stands. Therefore, we invite you to submit a revised version of the manuscript that addresses the points raised during the review process.

We look forward to receiving your revised manuscript.

Kind regards,

Mayank Sukhija

Academic Editor

PLOS ONE

Journal Requirements:

Additional Editor Comments:

Please address the comments of the Reviewer following that I would be happy to accept the manuscript for publication

Reviewers' comments:

Reviewer's Responses to Questions

**Comments to the Author**

1. If the authors have adequately addressed your comments raised in a previous round of review and you feel that this manuscript is now acceptable for publication, you may indicate that here to bypass the “Comments to the Author” section, enter your conflict of interest statement in the “Confidential to Editor” section, and submit your "Accept" recommendation.

Reviewer #1: All comments have been addressed

Reviewer #3: (No Response)

2. Is the manuscript technically sound, and do the data support the conclusions?

Reviewer #1: Yes

Reviewer #3: Yes

3. Has the statistical analysis been performed appropriately and rigorously? 

Reviewer #1: Yes

Reviewer #3: N/A

4. Have the authors made all data underlying the findings in their manuscript fully available?

Reviewer #1: Yes

Reviewer #3: No

5. Is the manuscript presented in an intelligible fashion and written in standard English?

Reviewer #1: Yes

Reviewer #3: Yes

6. Review Comments to the Author

Reviewer #1: (No Response)

Reviewer #3: Comment#3: The authors did not address the reviewer’s comment. The authors must state a specific value or range for “negative effect on low-temperature performance of the binder.” Instead of presenting some samples of articles, the authors must focus on addressing the reviewer’s comment properly.

7. PLOS authors have the option to publish the peer review history of their article (what does this mean? ). If published, this will include your full peer review and any attached files.

**Do you want your identity to be public for this peer review?** For information about this choice, including consent withdrawal, please see our Privacy Policy .

Reviewer #1: No

Reviewer #3: No

---

## [Author Response · Author response to Decision Letter 2]

8 May 2025

Explanation of the corrections that have been made by following the reviewers comments.

Many thanks for useful comments to our paper, and good advice for changes!

1. If the authors have adequately addressed your comments raised in a previous round of review and you feel that this manuscript is now acceptable for publication, you may indicate that here to bypass the “Comments to the Author” section, enter your conflict of interest statement in the “Confidential to Editor” section, and submit your "Accept" recommendation.

Reviewer #1: All comments have been addressed

We would like to thank the reviewer for his/her contributions and positive responses.

Reviewer #3: (No Response)

We would like to thank the reviewer for his/her contributions.

2. Is the manuscript technically sound, and do the data support the conclusions?

Reviewer #1: Yes

We would like to thank the reviewer for his/her contributions and positive responses.

Reviewer #3: Yes

We would like to thank the reviewer for his/her contributions and positive responses.

3. Has the statistical analysis been performed appropriately and rigorously?

Reviewer #1: Yes

We would like to thank the reviewer for his/her contributions and positive responses.

Reviewer #3: N/A

We would like to thank the reviewer for his/her contributions. In the manuscript, statistical analyses were conducted using the Statistical Package for the Social Sciences to evaluate the influence of various factors on the rutting parameter (G/sinδ) at a 0.05 significance level (95% confidence level). This analysis is presented in Section 4.4.1. We have observed that Reviewer 3 selected “Yes” for the same question in his/her previous review, as shown below:

“2. Has the statistical analysis been performed appropriately and rigorously?'

Reviewer #1: Yes

Reviewer #2: Yes

Reviewer #3: Yes”

For this reason, we believe that the “N/A” response in the current review may have been selected inadvertently.

4. Have the authors made all data underlying the findings in their manuscript fully available?

Reviewer #1: Yes

We would like to thank the reviewer for his/her contributions and positive responses.

Reviewer #3: No

We would like to thank the reviewer for his/her valuable comments. All data obtained from our study are presented in the tables and figures included in the manuscript. While the data in the tables are directly accessible, to ensure public accessibility and transparency, a minimal data set containing the underlying values used to generate the figures has been submitted as a supporting file titled “Minimal Data Set.”

5. Is the manuscript presented in an intelligible fashion and written in standard English?

Reviewer #1: Yes

We would like to thank the reviewer for his/her contributions and positive responses.

Reviewer #3: Yes

We would like to thank the reviewer for his/her contributions and positive responses.

6. Review Comments to the Author

Reviewer #1: (No Response)

We would like to thank the reviewer for his/her contributions.

Reviewer #3: Comment#3: The authors did not address the reviewer’s comment. The authors must state a specific value or range for “negative effect on low-temperature performance of the binder.” Instead of presenting some samples of articles, the authors must focus on addressing the reviewer’s comment properly.

We would like to thank the reviewer for his/her contributions. In response to the reviewer's comment, the last part of the Abstract has been revised as follows:

“…temperature of 300°C and rapid cooling. The low-temperature performance grade of biochar-modified asphalt binders decreased by one grade, while the high-temperature performance grade increased by at least one grade. The addition of biochar was shown to have a negative impact on the binder’s low-temperature performance. Based on these findings, it was concluded that the biochar additive, derived from industrial hemp stalk, is particularly well-suited for regions characterized by hot climates.”

With this, we hope that the article can be published in your Journal.

Kind regards.

---

## [Editor Report · Decision Letter 2]

Utilization of Biochar Derived from Industrial Hemp Stalks with Various Cooling Methods for Asphalt Binder Modification

PONE-D-25-02281R2

Dear Dr. Aslan,

We’re pleased to inform you that your manuscript has been judged scientifically suitable for publication and will be formally accepted for publication once it meets all outstanding technical requirements.

Kind regards,

Mayank Sukhija

Academic Editor

PLOS ONE
---

## [Editor Report · Acceptance letter]

PONE-D-25-02281R2

PLOS ONE

Dear Dr. Aslan,

I'm pleased to inform you that your manuscript has been deemed suitable for publication in PLOS ONE. Congratulations! Your manuscript is now being handed over to our production team.

Kind regards,

on behalf of

Dr. Mayank Sukhija

Academic Editor

PLOS ONE